# Molecular Docking Revealed the Potential Anti-Oxidative Stress Mechanism of the Walnut Polypeptide on HT22 Cells

**DOI:** 10.3390/foods12071554

**Published:** 2023-04-06

**Authors:** Zijie Zhang, Yuting Shang, Siting Li, Zhou Chen, Junxia Xia, Yiling Tian, Yingmin Jia, Aijin Ma

**Affiliations:** 1School of Food and Health, Beijing Technology and Business University, Beijing 100048, China; 2Hebei Yangyuan ZhiHui Beverage Co., Ltd., Hengshui 053000, China; 3College of Food Science and Technology, Hebei Agricultural University, Baoding 071000, China

**Keywords:** walnut meal, peptide, molecular docking, bioinformatics, antioxidant activity, cellular

## Abstract

The preparation of novel antioxidant peptides from food raw materials is one of the research focuses, but there are fewer studies on the preparation of antioxidant peptides from walnut meal, a by-product of processing walnuts. This study analyzed the antioxidant properties and protective effects of walnut protein hydrolyzed by alkaline protease and trypsin on the oxidative stress of HT22 cells. The peptides were identified by UPLC-MS/MS, and the anti-oxidative peptides were screened based on virtual computer tools. The potential anti-oxidative stress mechanism of the walnut polypeptide on HT22 cells was explored by molecular docking. The results revealed that walnut protein hydrolysates (WPH) with molecular weights of less than 1 kDa had good antioxidant properties and inhibited oxidative damage of HT22 cells by regulating the levels of reactive oxygen species (ROS) and antioxidant enzyme catalase (CAT), superoxide dismutase (SOD), and glutathione peroxidase (GSH-Px). Six of the ninety identified new peptides showed good solubility, non-toxicity, and bioactivity. The molecular docking results showed that the six peptides could dock with Keap1 successfully, and EYWNR and FQLPR (single-letter forms of peptide writing) could interact with the binding site of Nrf2 in the Keap1-Kelch structural domain through hydrogen bonds with strong binding forces. The results of this study provided important information on the antioxidant molecular mechanism of the walnut polypeptide and provided a basis for further development of walnut antioxidant polypeptide products.

## 1. Introduction

Walnuts are a nut with a wide global distribution. With its diverse and abundant walnut species, China is today the world’s top walnut producer [1]. Protein and fat in walnuts account for more than 70% of the weight of walnut kernels [2]. Walnut protein is mainly composed of eighteen amino acids, and its eight essential amino acids account for 31.82% of the total amino acid content. The FAO/WHO recommended that the intake of essential amino acids for adults can be fully satisfied by walnut protein, making it a high-quality plant protein resource [3]. Walnut peptides can be prepared by enzymatic and fermentation methods from walnuts or walnut meal from walnut oil extraction [4]. In terms of processing properties, walnut peptides have better solubility, hygroscopicity, emulsification, and foaming compared with walnut protein [5,6,7,8].

In addition, walnut peptides have a variety of biological functions, with significant effects on antioxidant properties, anti-inflammation, memory improvement, and Alzheimer’s disease risk reduction, as well as improving intestinal flora. Now, walnut peptides have become a research hotspot in the field of functional substances. Bioactive peptides of GGW, VYY, and LLPF sequences have been reported to be found in the hydrolysate of the defatted walnut meal, capable of improving the cause of learning and memory effects [9]. In addition, peptides such as TWLPLPR, YVLLPSPK, and KVPPLLY were found to have neuroprotective activity [10]. The potential mechanism of these peptides may be related to the ability to resist oxidative stress. Oxidative stress is a state in which the oxidative–antioxidative balance in the body is disturbed and oxidation becomes more intense, resulting in potential damage to the body [11]. Oxidative stress has been recognized as an important mechanism of damage in a number of diseases [12,13]. HT22 cells (mouse hippocampal neuronal cells) are derived from mouse hippocampal HT4 cells, an immortalized mouse hippocampal cell line. H_2_O_2_ molecules acting on HT22 cells are considered to be a common in vitro model of oxidative stress and are commonly used in antioxidant-related studies of active substances and drugs. Bioactive peptides can be used as natural antioxidants that are inexpensive and safe. Their inhibitory mechanisms have been studied in preliminary studies, making them ideal raw materials for the production of antioxidants.

Molecular docking has already helped to reduce the cost and time of activity mechanism studies by guiding experimental studies toward the best active substance more quickly than traditional research methods [14]. Molecular docking (MD) is one of the important methods of molecular simulation [15], which is essentially a recognition process between two or more molecules that involve spatial matching and energy matching between molecules. The implementation of this methodology permits the prediction of binding modes concerning peptides, small molecules, and ligands to their corresponding receptors, including enzymes. Employing molecular docking techniques, this research has successfully filtered antioxidant peptides while additionally identifying prospective pathways leading to antioxidant activity, thus substantiating the efficacy of such a technique [16]. Ultimately, the flexibility of molecular docking methods serves as an ideal tool for the detection of antioxidants and unraveling potential operational mechanisms.

The objective of this research was to separate and detect antioxidant peptides in walnut powder proteins using methods such as alkaline hydrolysis, ultrafiltration, LC-MS/MS identification, and computer simulation. Subsequently, employing virtual screening and molecular docking analyses, a molecular interaction model between the identified pentapeptides and the targeted molecules was proposed based on the results obtained.

## 2. Materials and Methods

### 2.1. Materials

Walnut meal provided by Hebei Yangyuan Zhihui Beverage Co., Ltd. (Hengshui, China). 1,1-diphenyl-2-picryl hydrazyl (DPPH), 2,2′-Azinobis-(3-ethylbenzthiazoline-6-sulphonate) (ABTS•+), Trypsin, and alkaline protease were purchased from Beijing Biotopped Technology Co. (Beijing, China). A BCA Protein Assay Kit (Biorigin, Louisville, KY, USA, BN16075) was purchased from Biorigin (Beijing) Inc. (Beijing, China). A Reactive Oxygen Species Assay Kit, a Catalase Assay Kit, and a Total Superoxide Dismutase Assay Kit with NBT were purchased from the Beyotime Institute of Biotechnology (Shanghai, China).

### 2.2. Preparation of Walnut Protein Hydrolysis Products

The walnut meal was produced in the process of pressing walnut oil production and was purchased from Hebei Yangyuan Zhihui Beverage Co., Ltd. (Hengshui, China). The walnut protein was separated by the alkali solubilization and acid precipitation method [17]. Walnut protein hydrolysate (WPH) was obtained by enzymatic hydrolysis, using the previous method with slight modifications [18]. A total of 100 g of walnut protein was added to 1.9 L of deionized water and stirred well using a magnetic stirrer. Them, adjust the pH to 8.0, add alkaline protease 400,000 U and trypsin 200,000 U, and hydrolyze at 50 °C for 4 h. Next, inactivate the enzymes in a water bath at 90 °C for 15 min. Finally, cool to room temperature, centrifuge at 4 °C, 8000 rpm for 15 min, take the supernatant, and freeze-dry. The lyophilized powder of walnut protein hydrolysis product was stored at −20 °C.

### 2.3. Ultrafiltration

WPH (5 mg/mL) was dissolved in ultrapure water and separated using ultrafiltration equipment (Millipore, MA, USA, working volume: 200 mL) at room temperature at 25 psi pressure, using 10 kDa, 3 kDa, and 1 kDa molecular weight ultrafiltration membranes (Millipore, MA, USA). The four fractions (F1: MW > 10 kDa, F2: MW 3–10 kDa, F3: MW 1–3 kDa, and F4: MW < 1 kDa) were sorted, lyophilized, and kept at −20 °C until further determination.

### 2.4. ABTS•+ Radical Scavenging Activity Assay

ABTS•+ was generated by mixing 0.35 mL of ABTS•+ diammonium salt (7.4 mM) with 0.35 mL of potassium persulfate (2.6 mM) to form an ABTS•+ working master batch, which was left to stand for 12–16 h away from light to form an ABTS•+ free radical stock solution. The ABTS•+ working solution was then diluted with 95% ethanol (approximately 1:35) to an absorbance of 0.70 ± 0.02 at 734 nm. A total of 160 μL of the ABTS•+ working solution was added to each well. A total of 40 μL of the sample was added to the sample wells, gently mixed, and incubated for 6 min at room temperature [19]. Absorbance values were measured at 734 nm. Ethanol instead of the sample was used as a blank, and ethanol instead of the ABTS•+ working solution was used as a control.
Radical scavenging activity (%) = (A_b_ − (A_s_ − A_c_))/A_b_ × 100,(1)
where A_b_ is the absorbance of the blank, A_s_ is the absorbance of the sample, and A_c_ is the absorbance of the control.

### 2.5. DPPH Radical Scavenging Activity Assay

A total of 100 μL of 0.1 mmol/L DPPH-anhydrous ethanol solution and 100 μL of different concentrations of the sample solution were added to the 96-well plate, mixed well, and then the absorbance value A_s_ was measured at 517 nm after incubated for 30 min in the dark at room temperature, and 100 μL of different concentrations of the sample solution was mixed with 100 μL of anhydrous ethanol, and the absorbance value A_c_ was measured. Ethanol was used as a blank instead of the sample solution, and the absorbance value A_b_ was measured [20].
DPPH radical scavenging activity (%) = (A_b_ − (A_s_ − A_c_))/A_b_ × 100,(2)
A_b_, A_s_, and A_c_ represent the absorbance of the blank, sample, and control, respectively.

### 2.6. Hydroxyl Radical (•OH) Scavenging Activity

•OH scavenging activity was measured using a kit (Suzhou Grace Biotechnology Co., Ltd., Suzhou, China). First, set up the blank group, control group, and sample group according to the instructions, and add the reagents sequentially. Set the time to 20 min (exact time) at 37 °C, transfer 200 μL to a 96-well plate, and read the absorbance value at 510 nm immediately.
Hydroxyl radical (•OH) scavenging activity (%) = (A_blank_ − (A_sample_ − A_control_))/A_blank_ × 100.(3)

### 2.7. Cell Culture and Cytotoxicity Assay

#### 2.7.1. Cell Culture

HT22 cells were cultivated in a DMEM comprehensive culture medium supplemented with 10% FBS and high sugar, and then maintained within a cell culture incubator featuring a 37 °C environment consisting of 5% CO_2_, 95% air, and 100% humidity. Inducing culture on either 96-well plates or 6-well plates at a density of 1 × 10^5^ cells per mL, cell growth was sustained for the duration of 24 h.

#### 2.7.2. Cytotoxicity Assay

An adapted version of the CCK-8 approach was employed to measure cytotoxicity levels in cells. The cells were sectioned into sets and processed similarly to the conditions detailed in 2.7.1, which included a blank control group and a sample group devoid of any treatment. Incubation proceeded within a CO_2_-enriched environment at 37 °C for a duration of 24 h. Following this, 100 μL of CCK-8 solution was carefully added to each well and incubated for an additional 2 h, after which an enzyme marker was utilized to assess the OD_450_ value.

### 2.8. Determination of ROS in HT22 Cells

An analysis of intracellular ROS using DCFH-DA as a fluorescent marker was performed [21]. Cells were grouped and treated according to the method in Section 2.7.1. The control (no treatment), model (1.0 mM H_2_O_2_), and sample groups (final concentrations of 0.1, 0.25, 0.5, and 1.0 mg/mL of WPH + 1.0 mM H_2_O_2_) were incubated in a 5% CO_2_ incubator at 37 °C for 24 h. The rest of the procedure was performed according to the requirements of the kit.

### 2.9. Determination of SOD, CAT, and GSH-Px in HT22 Cells

Cells were grouped and treated according to the method in 2.7.1. HT22 cells in good growth condition during the logarithmic growth period were inoculated in 6-well culture plates at 1.0 × 10^5^ cells/mL. The control (no treatment), model (1000 μM H_2_O_2_), and sample groups (final concentrations of 0.1, 0.25, 0.5, and 1.0 mg/mL of WPH + 1000 μM H_2_O_2_) were incubated in a 5% CO_2_ incubator at 37 °C for 24 h. The measurement of superoxide dismutase (SOD), catalase (CAT), and glutathione peroxidase (GSH-Px) activities were performed according to the kit (Nanjing Jiancheng Institute of Biological Engineering, Nanjing, China).

### 2.10. Identification of Peptide Sequences by LC-MS/MS

For the LC-MS/MS analysis, the desalted sample is loaded onto an Easy-nLC 1200 HPLC system (Thermo Fisher Scientific, Waltham, MA, USA) and then separated on an analytical column (C18,150 μm × 150 mm, 1.9 μm, Thermo Fisher Scientific, Waltham, MA, USA). The mobile phase A is 0.1% formic acid (*v*/*v*) in water and the mobile phase B is 20% 0.1% formic acid in water—80% acetonitrile. The total flow rate was fixed at 600 nL/min. MS scans (300–1800 *m*/*z*) were obtained in a Q-Exactive mass spectrometer (Thermo Fisher Scientific, Waltham, MA, USA) at a resolution of 70,000 at *m*/*z* 400. The raw MS files were analyzed and searched using PEAKS Studio. For downstream protein identification analysis, only highly confidently identified peptides were selected.

### 2.11. Computer Analysis of Identified Peptides

Using the PeptideRanker server (http://distilldeep.ucd.ie/PeptideRanker/ (accessed on 10 October 2022)), the potential biological activity of peptides identified by LC-MS/MS was predicted with reference to the previously described method. Peptides with a PeptideRanker score above 0.5 are predicted to be potentially biologically active and subjected to physicochemical properties by The ToxinPred tool (https://webs.iiitd.edu.in/raghava/toxinpred/index.html (accessed on 10 October 2022)) and In-novagen (http://www.innovagen.com/proteomics-tools (accessed on 10 October 2022)). The BIOPEP database (https://www.uwm.edu.pl/biochemia/index.php/en/biopep (accessed on 14 October 2022)) was used to examine the bioactive peptides obtained from LC-MS/MS.

### 2.12. Molecular Docking

Molecular docking studies were performed using the procedure in [22] to dock the identified peptide with Keap1 with some modifications. Prior to docking, the crystal structure of Keap1 (PDB ID: 2FUL) was used as the receptor, obtained by download from the PDB database. Meanwhile, the Keap1 structure was processed by PyMOL software (San Carlos, CA, USA) to remove water molecules and Nrf2 16-mer, adding hydrogen, minimizing energy, and removing excess ions and water molecules. The antioxidant peptide was used as a ligand. We used Chem Draw 20.0 software (PerkinElmer, Waltham, MA, USA) to draw the structure of peptides and AutoDock VINA software (The Center for Computational Structural Biology, La Jolla, CA, USA) to study the interaction between peptides and receptors. First, the root of the ligand is determined, the ligand’s twistable bond is selected, and the grid box is resized so as to wrap the active site of the receptor. Then, Running AutoDock VINA was used Next, we used Pymol to process the docking results. Interaction sites and 2D interaction maps of receptors and ligands were studied using the PLIP web tool (https://plip-tool.biotec.tu-dresden.de/plip-web/plip/index (accessed on 18 November 2022)) and proteins plus web tool (https://proteins.plus/ (accessed on 18 November 2022)), respectively.

### 2.13. Statistics Analysis

Based on three sets of parallel experiments, the data were expressed as −x ± s, and differences between groups were determined using a *t*-test. Statistical processing was performed using SPSS 22.0 (IBM SPSS Inc., Chicago, IL, USA) with significance and high significance considered when *p* < 0.05 and *p* < 0.01, respectively. GraphPad Prism 9.0(GraphPad Software, San Diego, CA, USA) was used to create graphs for visualization purposes.

## 3. Results

### 3.1. Chemical Antioxidant Activity of WPH Ultrafiltration Fractions

The antioxidant capacity of WPH was evaluated utilizing enzyme hydrolysis conditions that have been optimized based on pre-experimental results. (pH 8.0, temperature 50 °C, alkaline protease 400,000 U/g and trypsin 200,000 U/g, solid–liquid ratio 1:20 (g/mL.)) Ultrafiltration of the WPH through 10, 3, and 1 kDa filters produced four fractions of different molecular weights. The ABTS•+ (2,2-azino-bis (3-ethylbenzthiazoline-6-sulfonic acid))-based method is one of the widest measures of the total antioxidant capacity. The present investigation aimed to evaluate the antioxidant activity of varied molecular weights of WPH, using ABTS•+ scavenging activity detection methodology. In a 100 μg/mL concentration, comparing four fractions with different molecular weights, all fractions could effectively scavenge ABTS•+ radicals, and grade fraction F4 showed the highest free radical scavenging ability with 90.9 ± 1.7% ABTS•+ scavenging activity, and fraction F4 was selected for further experiments.

ABTS•+ scavenging activity, DPPH scavenging activity, and •OH scavenging activity are shown in Figure 1b–d. For ABTS•+ scavenging activity, the scavenging activity ranged from 10.1–98.8% in 12.5–200 μg/mL of walnut peptide in a dose-dependent manner to 98.2 ± 0.5%, and WPH showed strong ABTS•+ in this study’s scavenging ability. DPPH scavenging activity is considered one of the most standard and simple methods to assess the antioxidant properties of compounds [23], ranging from 36.7% to 89.1% in a dose-dependent manner in 25–200 μg/mL of walnut peptides. Hydroxyl radical is a kind of ROS with a strong and active oxidation ability. If not scavenged in time, it will cause damage and destruction to cells, tissues, and organs of the body and accelerate the aging of the body. Therefore, scavenging •OH can help to reduce ROS levels. In the 2.5–30.0 mg/mL walnut peptide, the scavenging activity ranged from 7.9% to 98.0% in a dose-dependent manner to 98.0 ± 1.6%. In a previous study, a plant-derived hydrolysis product prepared using neutral protease was found to have antioxidant activity with strong DPPH radical scavenging activity and ABTS•+ radical scavenging activity [24]. Similarly, animal-derived peptides with excellent antioxidant activity were studied and identified, with KAPDPGPGPM exhibiting the highest DPPH radical scavenging activity and peptide GGYDEY exhibiting ABTS•+ radical scavenging activity [25].

These results suggest that the small molecule WPH is an effective antioxidant. This is consistent with previous reports that hydrolysis products with smaller molecular weights have higher antioxidant activity [26]. As supported by research conducted previously, the lowest molecular weight fraction (<1 kDa) of walnut protein hydrolysis products displayed a markedly higher level of antioxidant activity. This observation may be attributed to the fact that small peptides, owing to their reduced molecular size, are capable of effectively and efficiently interacting with free radicals, even across the intestinal barrier [27]. However, it has been shown that the biological effect of antioxidants is not adequately reflected by the results of chemical methods [28]. Therefore, we further investigated the antioxidant activity of small molecule WPH at the cellular level.

### 3.2. The Cytoprotective Effect of WPH on H_2_O_2_-Induced Damage in HT22 Cells

The fundamental principle of this assay entails a reduction in WST-8 by dehydrogenase present in live cells, resulting in highly water-soluble orange-yellow chromogen generation. The intensity of the ensuing shade is proportional to the number of viable cells present. H_2_O_2_, as one of the major components of ROS, causes oxidative stress and leads to apoptosis or necrosis.

As shown in Figure 2a, the damage of H_2_O_2_ on HT22 cells showed a dose-dependent relationship. Compared with the blank group, the cell survival rate of HT22 cells decreased after 24 h of 500–1750 μM H_2_O_2_, respectively. Among them, 1000 μM of H_2_O_2_, was selected as the best half-lethal injury concentration for the oxidative stress model.

According to Figure 2, it can be observed that there was no substantial decrease in the viability of HT22 cells at concentrations ranging from 0.1 to 2 mg/mL for the fractions when compared to the blank group. These findings suggest that the small molecule WPH, at the aforementioned concentrations, does not possess cytotoxicity against HT22 cells. We further measured the ROS levels and antioxidant enzyme activity of HT22 cells.

### 3.3. The Effect of Small Molecule WPH Fractions on Intracellular ROS and Antioxidant Enzymes in HT22 Cells

The intracellular ROS content is an important marker to reflect the degree of oxidative damage in cells. In this study, DCFH-DA fluorescent dye was used to study the effect of ROS scavenging and the fluorescence intensity reflected the intracellular ROS level. As presented in Figure 2d, a meaningful increase in the relative fluorescence intensity of ROS was observed in the model group when compared to the blank group (*p* < 0.01), thereby indicating that the experimental model was effectively established. Interestingly, when the highest concentration of 1000 μg/mL was administered, there was a significant reduction in the relative fluorescence intensity of ROS, which was recorded as 112.34 RFU (*p* < 0.01) when compared to the model group.

Based on the data illustrated in Figure 3a–c, a significant decline was observed in the SOD, CAT, and GSH-Px activity within the model group when compared to the blank group (*p* < 0.01). Nevertheless, when pretreated with F4 at concentrations of 62.5–1000 μg/mL, significant improvements were noted in CAT, SOD activity, and GSH-Px content (*p* < 0.01) relative to the model group, which showed a dose-dependent relationship. This suggests that the F4 may scavenge some ROS through its antioxidant physiological function, reduce the damage of ROS on the cellular antioxidant enzyme system, maintain the normal operation of the cellular antioxidant system, and prevent further oxidative damage to cells. Similar results have been reported previously. Watermelon seed-derived antioxidant peptides significantly reduced ROS content in cells and significantly increased CAT, SOD, and GSH-Px activity [29].

### 3.4. Identification of Antioxidant Peptides

UPLC-MS/MS has demonstrated its effectiveness in the identification of amino acid sequences present in plant protein-derived peptides. In this study, a total of 90 amino acid sequences were successfully identified and characterized. The ToxinPred tool (https://webs.iiitd.edu.in/raghava/toxinpred/index.html (accessed on 10 October 2022)) and Innovagen (http://www.innovagen.com/proteomics-tools (accessed on 10 October 2022)) were utilized to evaluate the toxicity and physicochemical properties of the sample, which included the determination of potential isoelectric point (pI), charge, and theoretical grand average of hydropathicity (GRAVY). Table 1 lists their sequences, pI, charge, and theoretical GRAVY. A computer analysis of the above peptides was carried out to predict which sequences might be biologically active.

PeptideRanker is a server for predicting biologically active peptides. This server is a prediction tool for the probability that a peptide is biologically active. Any peptide predicted to exceed the 0.5 threshold was labeled as biologically active by PeptideRanker. Table 2 shows the top six peptides analyzed using PeptideRanker. Unreported peptides were obtained by comparing known antioxidant peptides in the BIOPEP-UWM database (https://biochemia.uwm.edu.pl/biopep-uwm/ (accessed on 14 October 2022)) and screening.

Studies have shown that the composition in terms of amino acids, sequence, and molecular structural features affects the antioxidant activity of bioactive peptides. Several previous studies have demonstrated that the number of amino acids present plays a crucial role in determining the antioxidant activity of peptides. Plant-derived peptides consist mainly of 2–13 amino acids and are able to exhibit excellent antioxidant capacity [30]. The number of peptides P1–P6 and amino acids in the six identified was 7, 5, 8, 6, 7, and 5, respectively, with satisfactory antioxidant capacity. The key antioxidant amino acid residues reported include hydrophobic amino acids such as Ala, Val, Leu, and Pro [31] and aromatic amino acids such as Phe, Tyr, and Trp [32]. Furthermore, the antioxidant properties of a peptide can also be influenced by the specific location of each amino acid residue. For instance, previous research suggests that the presence of a hydrophobic amino acid at either the N-terminus or the third position adjacent to the C-terminus can significantly impact the overall antioxidative potential of a peptide [33]. Additionally, studies have shown that including Tyr at the C-terminus may contribute considerably to the peptide’s antioxidant activity [30]. All six peptides screened exhibited these typical structural features.

### 3.5. A Molecular Docking Analysis of Keap1 and Peptides Derived from Walnut Proteins

Molecular docking is now used in a wide range of applications, such as predicting biological activity and studying the interaction of small molecule peptides and proteins [34]. The Keap1-Nrf2 pathway is a key pathway in the cellular oxidative stress response, and its regulated antioxidant proteins/enzymes play an important role in cellular defense protection, mainly including HO-1, peroxidase-1, SOD, GSH-Px, etc. [35]. Under physiological conditions, the overall intracellular environment is stable and most of Nrf2 exists in the cytosol in an inactive state, coupled with Keap1 [36]. As highlighted in this study, the Keap1 pocket appears to be particularly conducive to the small molecular binding. Specifically, it was found that certain small molecules are capable of readily binding to amino acid residues within this pocket, effectively occupying its active site and thereby inhibiting the ability of Keap1 to interact with Nrf2 [37].

Keap1 contains three functional domains, including a BTB structural domain, an IVR, and a DGR structural domain. the DGR structural domain can interact with the ETGE and DLG motifs of Nrf2, which is essential for maintaining the interaction between Nrf2 and Keap1. Direct disruption of the Keap1–Nrf2 interaction is thought to be a key target for the activation of this pathway [38]. Studies on Keap1 have shown that exposure to the ETGE motif of Nrf2 requires three Arg residues of Arg380, Arg415, and Arg483, four Ser residues of Ser363, Ser508, Ser555, and Ser602, and residues of Tyr334, Asn382, and Gln530 [39]. In addition, Tyr334, Arg380, Asn382, Arg415, Arg483, Tyr525, and Tyr572 contribute to the stability of the Kelch-Nrf2 complex [40]. All of the above residues are known as key residues of the Keap1-Kelch structural domain in the Keap1–Nrf2 interaction binding site.

It has been shown that bioactive peptides can disrupt Keap2–Nrf2 interactions by occupying the active site of Keap1 [41]. The 2D and 3D molecular interactions of P2 (a and c) and P6 (b and d) with the Keap1 active site, shown in Figure 4, were similar to the results for the antioxidant peptide of shredded rice protein origin, occupying the binding site of Nrf2 [42]. To more efficiently identify and analyze non-covalent interactions that may exist between Keap1 and P2 and P6, the Protein–Ligand Interaction Profiler (https://plip-tool.biotec.tu-dresden.de/plip-web/plip/index (accessed on 18 November 2022)) was employed. As shown in Table 3, Keap1 formed twelve hydrogen bonds with EYWNR, including six key residues (Arg380, Arg415, Arg483, Ser363, Ser555, and Ser602), while two hydrophobic forces contained one key residue (Arg415), and one salt bridge contained one key residue (Arg415). Similarly, as shown in Table 4, Keap1 forms eleven hydrogen bonds with FQLPR, including two key residues (Arg380, Ser555), while three hydrophobic interactions contain one key residue (Tyr334), and one salt bridge contains one key residue (Tyr334). Based on our experimental findings, it appears that EYWNR and FQLPR have the unique capability to occupy the Nrf2 binding site located in the Keap1-Kelch structural domain. By doing so, these peptides appear to directly inhibit the crucial Keap1–Nrf2 interaction, ultimately resulting in the release of free Nrf2. Thus, antioxidant peptides from walnut meal proteins have a good inhibitory effect on oxidative stress in neuronal cells.

## 4. Conclusions

The antioxidant activity of WPH and its antioxidant mechanism were investigated. Firstly, conventional antioxidant-guided graded isolation and molecular docking methods were used. The MW < 1 kDa hydrolysate of walnuts exhibited free radical scavenging and anti-oxidative stress protection, and two walnut antioxidant peptides were identified, EYWNR and FQLPR. Molecular docking studies showed that both peptides were able to interact spontaneously with Keap1 and block the entrance of its active site cavity, which causes cellular antioxidant enzymes to overexpress. In summary, EYWNR and FQLPR, two peptides with antioxidant activity, can be used as food-derived antioxidants for the development of functional foods or to prevent the oxidation of foods. In addition, the characteristic amino acids and potential mechanisms of peptide interactions with Keap1 were investigated. Further studies are needed to determine the cellular activity and in vivo effects of EYWNR and FQLPR in animal models. In future studies, we will further validate the results of walnut antioxidant peptides in animal experiments.

## Figures and Tables

**Figure 1 foods-12-01554-f001:**
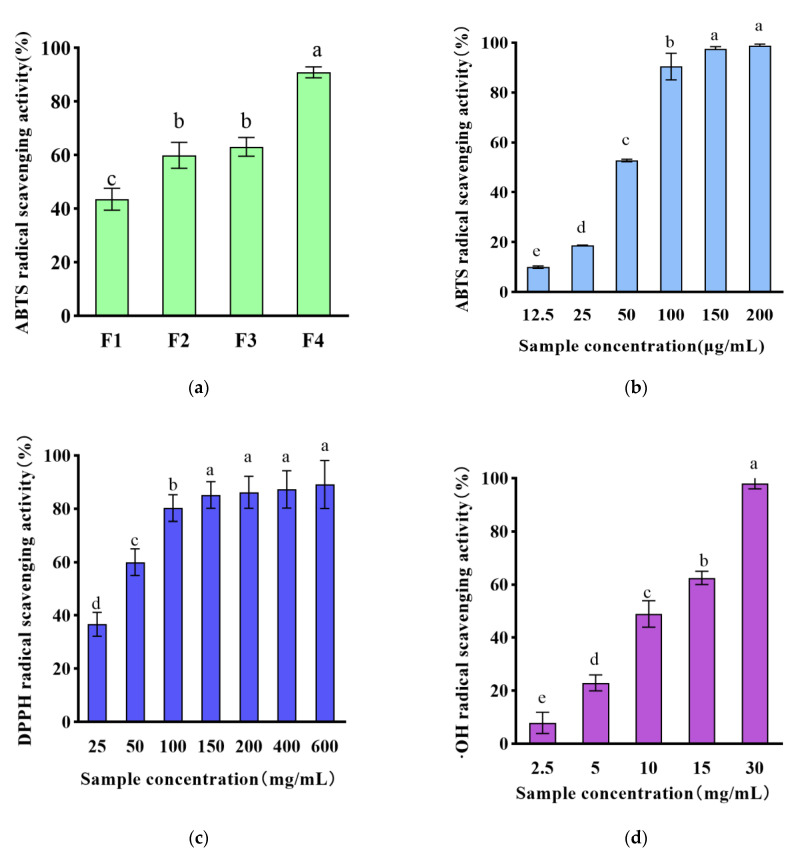
(**a**) Scavenging activity of ABTS•+ radicals of active fractions F1, F2, F3, and F4 obtained by ultrafiltration (%); (**b**) scavenging activity of ABTS•+ radicals of F4 fraction (%); (**c**) scavenging activity of DPPH radicals of F4 fraction (%); (**d**) scavenging activity of hydroxyl radicals of F4 fraction (%). The different lowercase letters at the top of the pattern bar in the picture represent significant differences between groups.

**Figure 2 foods-12-01554-f002:**
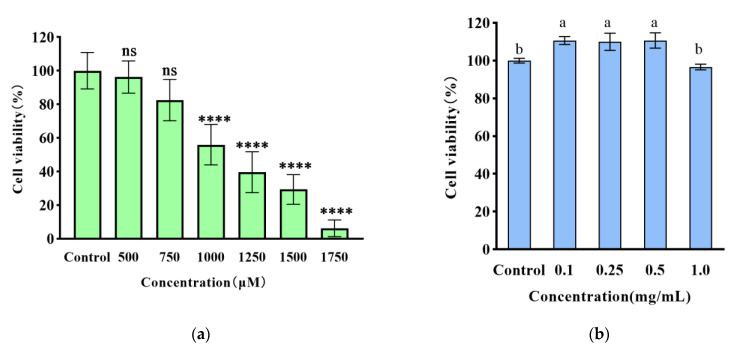
(**a**) The effect of H_2_O_2_ on the cellular activity of HT22 cells (%); (**b**) the effect of F4 on the cellular activity of HT22 cells (%); (**c**) the effect of F4 on cell viability of HT22 cells induced by H_2_O_2_ (%); (**d**) the effect of F4 on intracellular ROS production in HT22 cells. The different lowercase letters and asterisks at the top of the pattern bar in the picture represent significant differences between groups.

**Figure 3 foods-12-01554-f003:**
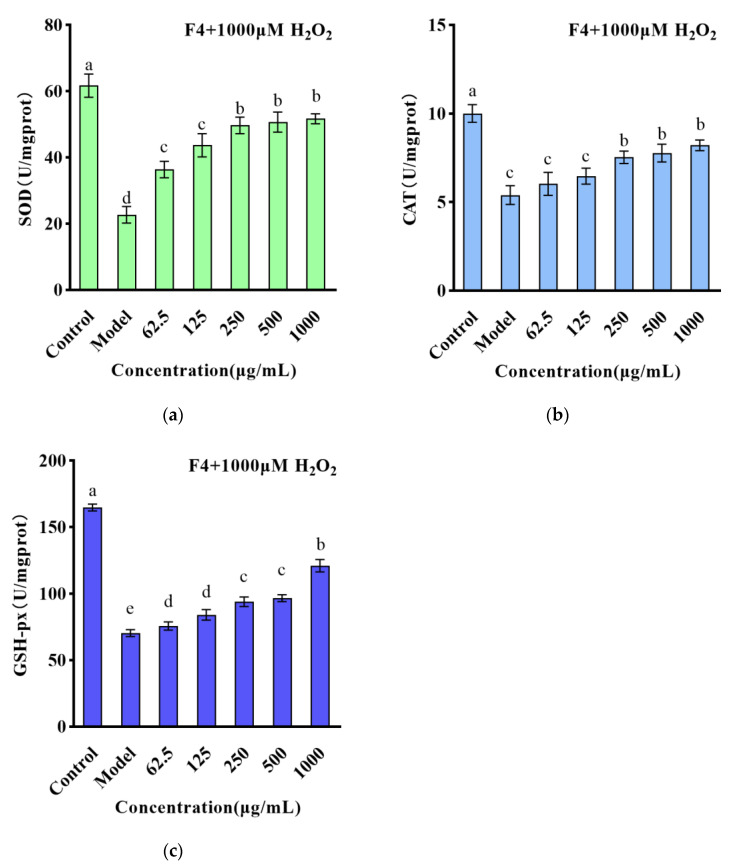
(**a**) The effect of F4 on the SOD content of HT22 cells induced by H_2_O_2_; (**b**) the effect of F4 on the CAT content of HT22 cells induced by H_2_O_2_; (**c**) the effect of F4 on the GSH-px content of HT22 cells induced by H_2_O_2_. The different lowercase letters at the top of the pattern bar in the picture represent significant differences between groups.

**Figure 4 foods-12-01554-f004:**
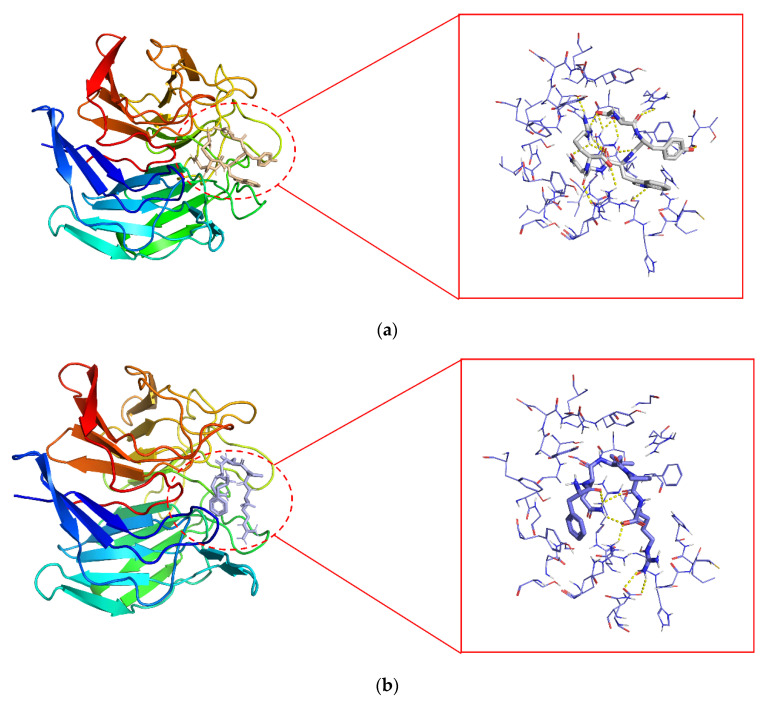
The 2D and 3D molecular interactions of P2 (**a**,**c**) and P6 (**b**,**d**) with the Keap1 active site (PDB ID: 6FLU). P2, EYWNR; p6, FQLPR.

**Table 1 foods-12-01554-t001:** Bioinformatic analysis of the identified peptides from the F4 fraction (MW < 1 kDa).

Sequence	Length	Isoelectric Point	Net Charge	GRAVY
VFDDELR	7	4.03	−2	−0.6
RVVDSEGGR	9	6.07	0	−1.022
TELLR	5	5.66	0	−0.22
LLSSSR	6	9.75	1	0.117
VDHLR	5	6.71	0	−0.64
PDTLAR	6	6.27	0	−0.783
YVTLK	5	8.59	1	0.42
FGNVLTK	7	8.75	1	0.329
TVEDELR	7	4.14	−2	−1.1
ESWPGSR	7	6.1	0	−1.786
VLRPR	5	12	2	−0.52
EYWNR	5	6.1	0	−2.74
LAGNPHQQ	8	6.74	0	−1.262
VYDDELR	7	4.03	−2	−1.186
SDFVSR	6	5.55	0	−0.433
AGVDLVR	7	5.88	0	0.8
LLPEER	6	4.53	−1	−0.917
VVAPEPR	7	5.97	0	−0.143
SSTDRLR	7	9.31	−1	−1.571
LPDTLAR	7	5.84	0	−0.129
VPGLLEKEP	9	4.53	−1	−0.3
LLSSSR	6	9.75	1	0.117
RTLEPTR	7	9.6	1	−1.671
VFDGELR	7	4.37	−1	−0.157
SPDQSYLR	8	5.55	0	−1.525
YAELKR	6	8.59	1	−1.267
VVLLR	5	9.72	1	2.3
SNHANQLDR	9	6.46	0	−1.878
KLPLLR	6	11	2	0.233
GVSEDKLQR	9	6.07	0	−1.344
KVVVPK	6	10	2	0.533
LDVLR	5	5.84	0	0.76
VEDELR	6	4.14	−2	−1.167
KLPLLR	6	11	2	0.233
RFEEERQR	8	6.23	0	−3.087
TESDVFR	7	4.37	−1	−0.857
SQRPDLQPR	9	9.31	1	−2.189
NDSTHPF	7	5.08	−1	−1.5
LLPVLR	6	9.75	1	1.583
FQLPR	5	9.75	1	−0.6
LTVEDELR	8	4.14	−2	−0.487
HLVGPDK	7	6.74	0	−0.657
LPDTLAR	7	5.84	0	−0.129
FDVGVK	6	5.84	0	0.567
YFHSQ	5	6.74	0	−1.2
EVLRPR	6	9.7	1	−1.017
VEDELRVL	8	4.14	−2	0.125
LLAGGERP	8	6	0	−0.125
VFDDELER	8	3.92	−3	−0.963
SHSVLYR	7	8.49	1	−0.371
AVDNAYAR	8	5.88	0	−0.4
YSNAPR	6	8.75	1	−1.65
FADLTK	6	5.84	0	0.05
YVPTER	6	6	0	−1.233
SAGQRPW	7	9.47	1	−1.414
TFLAR	5	9.41	1	0.64
YVTLRK	6	9.99	2	−0.4
HADLPGLK	8	6.74	0	−0.4
VFDDDRLR	8	4.43	−1	−1.087
TLPVLR	6	9.41	1	0.833
LNTPR	5	9.75	1	−1.3
SEDKLQR	7	5.79	0	−2.271
LPLLR	5	9.75	1	1.06
VDLVR	5	5.81	0	0.84
SSGGPLSLR	9	9.47	1	−0.189
EGVLLRK	7	8.85	1	−0.071
TNSFQMSPR	9	9.41	1	−1.189
VYLAR	5	9.81	1	1.06
SDVFSR	6	5.55	0	−0.433
VSLPKP	6	8.72	1	0.017
YGNVLR	6	8.75	1	−0.283
FDGELR	6	4.37	−1	−0.883
SNDLLR	6	5.55	0	−0.783
TNRPQF	6	9.41	1	−1.833
LDREERAR	8	6.18	0	−2.3
LVHPQR	6	9.76	1	−0.8
FLDLLK	6	5.84	0	1.133
MPLDVLR	7	5.59	0	0.586
YAELK	5	6	0	−0.62
GVDLVKR	7	8.75	1	−0.014
TESDVFSR	8	4.37	−1	−0.85
VDAHPLR	7	6.71	0	−0.429
LTPVLR	6	9.75	1	0.833
SPWER	5	5.72	0	−2.26
ASFQR	5	9.79	1	−0.84
AFNFPAR	7	9.79	1	−0.057
SLVRVE	6	5.72	0	0.567
VSPDLVR	7	5.81	0	0.257
TDEYGNVLR	9	4.37	−1	−1.044
TEQAGRLS	8	5.66	0	−0.975

**Table 2 foods-12-01554-t002:** Spectral intensity, predicted toxicity, solubility, and bioactivity scores of selected peptides.

Fraction No.	Sequence	Area	Toxin	Estimated Solubility	PeptideRanker Score
P1	ESWPGSR	1.37 × 10^8^	Non-Toxin	Good	0.614027
P2	EYWNR	6.17 × 10^7^	Non-Toxin	Good	0.516512
P3	SPDQSYLR	3.77 × 10^7^	Non-Toxin	Good	0.609955
P4	KLPLLR	3.08 × 10^7^	Non-Toxin	Good	0.548661
P5	NDSTHPF	2.42 × 10^7^	Non-Toxin	Good	0.549226
P6	FQLPR	2.21 × 10^7^	Non-Toxin	Good	0.812155

**Table 3 foods-12-01554-t003:** Interaction force and the P2 binding site with the Keap1 active site (PDB ID: 2FLU).

Intermolecular Forces	Index	Residue	AA
Hydrophobic interaction	1	415X	ARG
	2	525X	TYR
Hydrogen bonds	1	363X	SER
	2	380X	ARG
	3	380X	ARG
	4	415X	ARG
	5	415X	ARG
	6	431X	SER
	7	431X	SER
	8	480X	GLY
	9	483X	ARG
	10	555X	SER
	11	555X	SER
	12	602X	SER
π-stacking	1	478X	PHE
Salt bridge	1	415X	ARG

**Table 4 foods-12-01554-t004:** Interaction force and the P6 binding site with the Keap1 active site (PDB ID: 2FLU).

Intermolecular Forces	Index	Residue	AA
Hydrophobic interaction	1	334X	TYR
	2	525X	TYR
	3	556X	ALA
Hydrogen bonds	1	380X	ARG
	2	380X	ARG
	3	414X	ASN
	4	431X	SER
	5	431X	SER
	6	433X	GLY
	7	433X	GLY
	8	530X	GLN
	9	555X	SER
	10	555X	SER
	11	555X	SER
π-stacking	1	334X	TYR

## Data Availability

Data is contained within the article.

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
