# Peer review of "Molecular Docking Revealed the Potential Anti-Oxidative Stress Mechanism of the Walnut Polypeptide on HT22 Cells"

_foods, 2023, doi:10.3390/foods12071554_

Round 1
Reviewer 1 Report
1. Since HT22 cells were important media to evaluate antioxidant activities, the cells should be reviewed and described in the “Introduction”.
2. The “100% “ should be changed to “100” in Equation (1), (2) and (3)
3. Ultrafiltration membranes were not described clearly. There was no brand, working volume and operation procedure related to the membranes.
4. The method for ABTS•+ Radical scavenging activity was lack of “Control”. The control should be obtained by mixing 0.35 mL of diammonium salt (7.4 mM) instead of ABTS•+ diammonium salt (7.4 mM). Equation (1) did not involve the “Control”.
5. In equation (2), distilled water was used as a blank control instead of the sample solution. However, the whole system to determine DPPH was prepared in a methanol solution. Methanol should be applied as a blank control.
6. Peptides P1-P6 (line 287) should be denoted on Table 2.
Reviewer 2 Report
This paper performs in a good form but many points are needed to revised:
1. English mistakes and error typing, can be found in many places of the paper, for instance, line 22, "in" should be separated. Line 182, chemical should be "Chemical"
2. References should be modified following the style of Foods in MDPI, journal names must be abbreviated.
3. Many abbreviations should be explained, for instance in the Abstract, CAT, SOD...should be explained. Similarly, inline 45, peptide names should be explained.
4. The similarity is 32%, should be reduced. If they can reduce <20%, will be a very good paper.
5. Line 177, should be follow the style of MDPI
6. t-test, t should be in italic, P should be p and in italic (lines 179-180). Please do in all other places of the paper.
7. In their Figures, for instance Figure 1, letters should be in consistent with letters of the manuscript. They used Time News Roman.
8. Line 248, after Figure 2(d), a space must be deleted. Figure 2 should be capitalized.
9. Line 252, in vivo should be italic
I can not check all places, please carefully read and revise your paper again.
Reviewer 3 Report
In this manuscript, You have shown (in vitro) that low molecular weight protein hydrolysates prepared from walnut meal have good antioxidant properties and exhibited free radical scavenging activity. Also, in silico molecular docking results have shown the potential mechanisms of peptide interactions with Keap1. The manuscript is well written, however, I have some questions regarding your methodology and conclusions.
1. In section 2.12 about molecular docking procedure, what are the modifications regarding the procedure you referenced (ref. 22). Also, what are the other necessary procedures you mentioned in line 176? Since the manuscript title is „Molecular docking revealed the potential anti-oxidative stress mechanism of walnut polypeptide on HT22 cells“ I deem it necessary to describe molecular docking procedure in more detail.
2. It is well known that amino acids, di- and tri-peptides are taken up from the intestinal ileum into enterocytes through various transporters (such as PepT1). Since both peptides mentioned in the conclusion are pentapeptides, can you substantiate your claim that they could be used in the development of functional foods? Will they protect the food itself from oxidation or are they supposed to protect us (prevent oxidation in our cells) upon this food consumption?
Round 2
Reviewer 1 Report
No further comments.